# Entrepreneurial, Economic, and Social Well-Being Outcomes from an RCT of a Youth Entrepreneurship Education Intervention among Native American Adolescents

**DOI:** 10.3390/ijerph17072383

**Published:** 2020-03-31

**Authors:** Lauren Tingey, Francene Larzelere, Novalene Goklish, Summer Rosenstock, Larissa Jennings Mayo-Wilson, Elliott Pablo, Warren Goklish, Ryan Grass, Feather Sprengeler, Sean Parker, Allison Ingalls, Mariddie Craig, Allison Barlow

**Affiliations:** 1Department of International Health, Johns Hopkins Center for American Indian Health, 415 N. Washington St., Baltimore, MD 21231, USA; summer.rosenstock@jhu.edu (S.R.); aingalls@jhu.edu (A.I.); abarlow@jhu.edu (A.B.); 2Department of International Health, Johns Hopkins Center for American Indian Health, 308 Kuper St., Whiteriver, AZ 85941, USA; flarzel1@jhu.edu (F.L.); ngoklis1@jhu.edu (N.G.); epablo1@jhu.edu (E.P.); warrengoklish51@gmail.com (W.G.); rgrass1@jhu.edu (R.G.); fsprengeler@firstthingsfirst.org (F.S.); seanduaneparker1@gmail.com (S.P.); mariddie.ann@gmail.com (M.C.); 3Department of Applied Health Science, Center for Sexual Health Promotion, Indiana University School of Public Health, 1025 E. 7th St., Bloomington, IN 47405, USA; ljmayowi@iu.edu

**Keywords:** Native American, adolescent, entrepreneurship education, randomized controlled trial, economic empowerment, qualitative

## Abstract

*Background:* Entrepreneurship education has demonstrated positive impacts in low-resource contexts. However, there is limited evidence of such programs evaluated among Native American (NA) youth in a rural reservation. *Methods:* A 2:1 randomized controlled trial evaluated the impact of the Arrowhead Business Group (ABG) entrepreneurship education program on entrepreneurship knowledge, economic empowerment, and social well-being among 394 NA youth. An intent to treat analysis using mixed effects regression models examined within and between study group differences from baseline to 24 months. An interaction term measured change in the intervention relative to change in the control. ABG participants were purposively sampled to conduct focus groups and in-depth interviews. *Results:* Significant intervention vs. control group improvements were sustained at 12 months for entrepreneurship knowledge and economic confidence/security. Significant within-group improvements were sustained for ABG participants at 24 months for connectedness to parents, school, and awareness of connectedness. Qualitative data endorses positive impacts on social well-being among ABG participants. *Conclusion:* Observed effects on entrepreneurship knowledge, economic empowerment, and connectedness, supplemented by the experiences and changes as described by the youth themselves, demonstrates how a strength-based youth entrepreneurship intervention focused on developing assets and resources may be an innovative approach to dually address health and economic disparities endured in Native American communities.

## 1. Introduction

The winners of the 2019 Nobel Memorial Prize in Economic Sciences, Drs. Abhijit Banerjee, Esther Duflo, and Michael Kremer, are touted for advancing development economics methodology in which they run real-world controlled trials in small localities to gather evidence on how to reduce global poverty [1,2,3]. Another hallmark of their work is a departure from narrow finance-focused observation and analytics, to the inclusion of health and educational indicators. This paper is consistent with their award-winning approach and reports on a real-world controlled trial completed in a small locality evaluating the impact of an intervention on health and educational outcomes with the long-term goal of reducing poverty. More specifically, this paper describes entrepreneurship, economic and social well-being outcomes from the first randomized controlled trial (RCT) of a culturally based youth entrepreneurship education intervention among Native American youth living on the Fort Apache Indian Reservation in rural Arizona.

Participating youth, who are members of the White Mountain Apache Tribe (Apache), are exposed to high rates of poverty, high school drop-out, suicide, and substance use, as well as generational and modern traumas [4,5,6]. They are also part of a resilient and innovative peoples who have used research for over 40 years to develop public health solutions that have scaled to the world [7]. A recent paper from this trial reported on positive mental and behavioral outcomes among intervention participants—including less marijuana use, binge alcohol use, suicide attempts, physical fighting, and fights at school, compared to same-age peers who did not receive the intervention [8]. This paper will report on intervention impacts among participating youth’s entrepreneurship, economic, and social well-being outcomes.

### 1.1. Evidence from Past Youth Entrepreneurship Trials

Past evidence has identified youth entrepreneurship education as a promising, strength-based, protective factor model [9,10,11,12,13,14]. Entrepreneurship education interventions generally employ a positive youth development approach to cultivate assets and resources to bolster the social well-being and potential of adolescent youth exposed to inordinate behavioral and economic risks [15,16,17,18,19,20]. To accomplish this, entrepreneurship education targets internal assets, such as entrepreneurship knowledge, economic confidence and security, and future planning, as well as resources, such as connectedness to school; relationships with parents, teachers, and other role models; and participation in productive economic activities [16,17,18]. Entrepreneurship education interventions represent a significant and meaningful shift from a focus on “fixing” adolescents, to embracing youth’s strengths and capacity for making healthy choices [18,21].

While very few entrepreneurship education interventions have been implemented on Native American reservations and none until now have been rigorously evaluated, entrepreneurship education interventions appear particularly promising for Native American (NA) youth living on reservations for two reasons: (1) Strengths-based approaches are highly congruent with Native views of ways to promote health and well-being [22,23,24,25,26,27,28,29,30], and (2) NA adolescents face some of the worst social, economic, health, and education risks of any racial or ethnic group in the U.S. [31,32,33,34]. Increasing recognition that social determinants, most notably poverty, and health status are inextricably linked supports the need for rigorous research on positive youth development approaches addressing both [1,2,3].

A review of NA adolescent research yielded numerous protective factors for economic and social well-being in Native populations that are consistent with targets of entrepreneurship education, including: Belonging, cultural connectedness, and positive relationships with caring adults, leaders, and role models; mastery, independence, and self-efficacy; positive attachment to school, as well as hope and optimism [24,25,27,29,35,36,37,38,39,40]. However, given the many obstacles to job creation, low employment rates, structural and historical barriers to economic mobility, and potential differences in economic value systems among Native peoples, [2,3,9,21] an important focus of our work is to understand how Native adolescents understand and derive meaning from the entrepreneurship program content and potential positive connections to peers and adults.

### 1.2. The Purpose of This Paper

The purpose of this paper is to report on entrepreneurship, economic, and social well-being outcomes from the 5-year RCT of ABG (#NCT02157493; clinicaltrials.gov). The original aims of the study were to longitudinally evaluate ABG for impacts on: (1) Adolescent behavioral and mental health outcomes, including substance use, suicide, and violence to self and others; (2) social well-being; (3) educational; and (4) entrepreneurship/economic outcomes [41,42]. This paper focuses on outcomes from aims 2 and 4. Behavioral and mental health outcomes (aim 1) have already been reported [8] and educational outcomes (aim 3) are still being extracted from participating youth’s academic records.

## 2. Materials and Methods

### 2.1. The Arrowhead Business Group Youth Entrepreneurship Model

The ABG intervention and evaluation design was developed over a two-year formative period by the White Mountain Apache Tribe (WMAT/Apache) in collaboration with Johns Hopkins University (JHU) Center for American Indian Health. Described in more detail elsewhere, ABG consists of 16 lessons delivered through a residential summer camp followed by 6 monthly follow-on workshops, 4 to 6 h in length, to develop business plans [41,42]. Lessons and activities are taught by two Apache paraprofessional facilitators to groups of 25 youth participants, aged 13 to 16 years old. ABG content—in successive order—includes: Apache culture and history and historical examples of entrepreneurship; problem solving and coping skills; communication, decision making, and goal setting; financial literacy, entrepreneurship training, and small business design; marketing; and development. The initial emphasis on Apache historical survival and cultural values, followed by soft skills (i.e., communication, problem-solving, and coping skills), before introducing financial and entrepreneurship training, was decided through the community-based development of the program. ABG lessons are supplemented with presentations by Apache entrepreneurs and elders to further promote culturally congruent concepts of entrepreneurship and connectedness to Apache culture and identity.

A community advisory board (CAB) and Apache program staff recommended the importance of cultural grounding and soft skills development before delivering the more didactic entrepreneurship training, which they thought youth may perceive as irrelevant or unattainable without the introductory components. In addition, the format of the ABG curriculum was explicitly intended to support connections to school—as same-school peers participate together in ABG camps and content provides rationale for staying in school—in addition to connections to cultural and community leaders who meet with youth throughout the ABG intervention period. At the end of the program, participants present business ideas to a review board comprised of local business leaders and can receive start-up funds and mentorship from ABG program staff.

### 2.2. A Peripheral Asset: The Arrowhead Café and Marketplace

Originally, the six monthly ABG follow-on workshops were taught in a small conference room at historic Fort Apache, inside an early 20th century structure once occupied by the U.S. Calvary to suppress the Apache people. However, after the ABG program and evaluation began, local program leaders were inspired to launch the ABG Café and Marketplace with private funds raised for this purpose next to where the ABG workshops were taking place. Once the ABG Café and Marketplace were opened, program participants were able to seek apprenticeship opportunities there, and could borrow production supplies from an equipment co-op that includes cameras, screen-printing equipment, jewelry making materials, and art supplies. Over the course of this study, the Café and Marketplace, staffed with ABG program participants, grew into a popular eating and meeting place that also sold and promoted business products or services created by ABG youth participants and community members alike. The Café and Marketplace was inclusive and open to everyone in the community. Therefore, youth who were not participating in the ABG program also took advantage of the Café and Marketplace’s services. Involvement with the Café and Marketplace by youth who were not assigned to the ABG intervention may have potentially affected some of the trial outcomes. This positive unintended consequence of the ABG program is further elaborated in the discussion.

### 2.3. Study Design

We used a 2:1 RCT study design to evaluate ABG for impacts on key quantitative social well-being, entrepreneurial, and economic outcomes. We also conducted focus groups and in-depth interviews to collect qualitative data with a subset of youth who participated in ABG. A detailed description of the study rationale, methods including a CONSORT diagram illustrating the study flow, and baseline characteristics of participants may be found elsewhere [41,42]. The study design was reviewed and approved by the Apache Tribal Council and Health Advisory Board, and the JHU Institutional Review Board (IRB) (#5616). This manuscript was reviewed and approved by the Tribal Council and Health Advisory Board. The study was conducted from May 2014 to June 2019.

### 2.4. Participants

Participants were ages 13–16, NA (self-identified)—94% Apache, 3% Navajo, 2% San Carlos Apache, and 1% other Native American ethnicity—and living on the Fort Apache Indian Reservation. Local study staff obtained informed consent from a parent or legal guardian for the adolescent’s participation and assent from minor participants. We partnered with 11 area middle and high schools for recruitment; youth were recruited in three annual cohorts. In each cohort, participants were enrolled by the study team, randomized, received their designated program from a trained facilitator, and completed self-report follow-up surveys out to 24 months post-intervention. Participants were randomized 2:1 through a randomization sequence generated via software by the study’s statistician in block sizes of 6, 9, or 12 to receive the control condition (which consisted of three 4- to 5-h recreational sports field days held at various popular outdoor locations on the reservation) or the ABG intervention plus control. This study design was endorsed by Apache research staff and our local CAB and ensured differences on key outcomes of interest could be attributed to receipt of the ABG intervention. Neither research staff nor study participants were blinded to randomization status.

### 2.5. Quantitative Outcomes

We collected data at baseline and at 6-, 12-, and 24-months post-intervention through self-report questionnaires on hard copy or through Audio Computer Assisted Self-Interview technology (ACASI) on a laptop or tablet. These two methods of data collection were utilized to ensure valid real-time data collection in a rural reservation-based context. Entrepreneurship knowledge and economic empowerment questions were developed by the study team and based on the extant entrepreneurship and economic empowerment literature (additional detail provided below). Social well-being outcomes included connectedness measured by the Hemingway Measure of Adolescent Connectedness, and the Awareness of Connectedness Scale; and hopelessness measured by the Hopelessness Scale, and hopefulness measured by the Apache Hopefulness Scale [43,44,45,46]. All questions were reviewed by the local Apache study team, piloted with Apache youth, and revised as necessary prior to utilization. See Table 1 for a complete description of the quantitative outcome variables included in this analysis.

Entrepreneurship knowledge variables were designed to reflect key content of the ABG curriculum; response options were true/false and multiple choice. Entrepreneurship knowledge questions were organized by the following subscales with illustrative examples. *Subscale A: Personal Finance and Business Planning:* Subscale A includes 18 items, with questions primarily focused on personal financial/business planning knowledge and well-being. Example questions include: What daily expense can you give up in order to save for a big item, such as a laptop? When it is best to start a savings account? *Subscale B: Entrepreneurial Knowledge and Business Development:* Subscale B includes 13 items with questions addressing entrepreneurial/business development knowledge. Example questions include: What types of businesses must rely on donations to pay for expenses? What does a target market include? A sum of the correctly answered items was calculated to form a total knowledge index, inclusive of questions from index A and B. In addition, separate sums were calculated for index A and index B.

Economic empowerment variables were answered on a Likert scale (1 = Strongly Agree; 4 = Strongly Disagree). Likert scale questions were coded 1—strongly disagree to 4—strongly agree. Several questions required reverse coding to align with the direction of the other questions included. A mean score across items was calculated for each economic empowerment domain. Economic empowerment variables were organized by the following subscales:

*Expansion of Current Economic Abilities (11 questions)* [47,48,49,50]. I work for pay either as cash or in kind, such as food or housing; I have a talent or skill that can be used to earn money; I have supplies I can use to earn money; I currently have cash savings; I can buy things (clothes, food, supplies) that my family needs; I am becoming financially stronger each day; I am learning how to manage my finances; I am learning how to earn income for myself and my family; I have new opportunities to earn income; I have people to help me earn income; I can take part now in more activities to earn income.

*Economic Agency and Participation (9 questions)* [49]. My family talks to me about how money in our household is spent; I alone decide how the money I earn will be used; I often spend the money I earn on myself; I often spend the money I earn on others; I am a part of my family’s income source; I am an important member of the household; others listen to what I have to say; I have a say in what is best for my well-being; I have money to support what makes me happy.

*Economic Confidence and Security (7 questions)* [49]. My family values the income I provide; I can ask or bargain for what is important to me; I can help my family with unexpected and large expenses; I feel the family’s stress of having little money; I worry about not having enough money to survive; there is something I can do to make my family’s financial future stronger; I am well able to take care of myself and my family.

*Future Planning and Aspirations (11 questions)* [16,19,20,51,52,53]. I have a plan for how I want my life to be. When I grow up, I will follow my interest where they take me; I am preparing now for the full life I want to live; when I grow up, I am certain I will have enough money to buy the things I need; thinking about the future makes me feel happy; I can make new plans now that I have a source of income; I feel inspired to make a better life for myself; I want to do something my family will feel proud of; this past year I have started a new goal; I feel I have much to be proud of; I want to teach other young people to be like me.

*Intentions to Preserve Health (14 questions)* [16,19,51,52]. I think a lot about my well-being; I work hard to remain healthy, even when others around me become ill; I avoid situations that will harm my health and well-being; if it is possible I will get sick, I am diligent not to let it happen; I believe one should never give up; I occasionally take risks that could be harmful to my health; sometimes, I worry about my health; there is nothing I can do to remain healthy; there are some things I can do to avoid getting sick; I am more careful than before to ensure that I am healthy and strong; I do not always protect my health the way I should; I am well capable of avoiding an illness or infection; it is important to me that I live a long and healthy life; I will say no to others if I am in danger.

Connectedness. Several subscales of the Hemmingway Measure of Adolescent Connectedness were included in this analysis, including connectedness to parents (6 Likert scale item—range 1–5, alpha = 0.7223), mother (5 Likert scale items—range 1–5, alpha = 0.7414), father (5 Likert scale items—range 1–5, alpha = 0.7767), school (6 Likert scale items—range 1–5, alpha = 0.8023), and teachers (6 Likert scale items—range 1–5, alpha = 0.7337). Means were calculated for each subscale and ranged from 1 to 5, with higher scores indicating greater connectedness.

Awareness of Connectedness. The sum was calculated of the items included in the Awareness of Connectedness scale and ranged from 0 to 60, with higher scores indicating higher awareness of connectedness (12 Likert scale items—range 0–5, alpha = 0.8434).

Hopelessness. The Hopelessness Scale is a sum of 16 items with a range of 1–16, with higher scores indicating higher levels of hopelessness (16 true/false items—range 0–1, alpha = 0.6982).

Hopefulness. The Apache Hopefulness Scale was calculated as the mean across items with a range of 1–5, with higher scores indicating higher levels of hopefulness (13 Likert scale items—range 1–5, alpha = 0.8149).

### 2.6. Qualitative Outcomes

To complement and extend the quantitative data, we conducted focus group discussions and qualitative interviews with a subset of youth who participated in the ABG intervention to better understand how they understood and digested curriculum content, and potential mechanisms of intervention-related change. Participants were purposively sampled after completion of the ABG program. We conducted two focus groups with mixed-sex groups of youth and eight in-depth interviews with individual youth. Focus groups and in-depth interviews were conducted at our local study office by Apache research staff using semi-structured interview guides. Interview guides asked participants what they liked or disliked about the ABG program, how participating in the program has changed their lives, how their relationships have changed as a result of participating in the program, how they view Apache culture, and specifically their opinions regarding Apache values around entrepreneurship. Qualitative data were collected to enhance understanding of how the program was being received by the participants in terms of their economic, entrepreneurship, and social well-being outcomes. We did not have resources to reach data saturation but were able to gather consistent rich illustrative quotes from the voices of participating Apache youth.

### 2.7. Quantitative Analysis

Stata 14 was used to conduct all quantitative statistical analyses [54]. Demographic data and quantitative outcomes of interest were tested at baseline for between-group comparability using t-tests for continuous variables and chi^2^ tests for categorical variables. Intent to treat analysis (ITT) was used to examine between study group differences (intervention vs. control) in entrepreneurship knowledge, including both subscales, the economic empowerment scale inclusive of five domains, and the aforementioned social well-being outcomes.

We used linear and logistic mixed effects regression models to examine within and between study group differences from baseline to 24 months follow-up for key economic, entrepreneurship, and social well-being outcomes. Regression models included study group, time point, and an interaction term between study group and time point. An interaction term was used to examine change in the intervention relative to change in the control group. We adjusted all models for repeated measures.

### 2.8. Qualitative Analysis

We audiotaped and transcribed focus groups and interviews verbatim. Two independent coders manually coded the transcripts. We used a descriptive method of qualitative content analysis to further elaborate the quantitative outcomes of the ABG program [55]. With a descriptive method of content analysis, we were able to gather quotes exemplifying youth’s own interpretation of how the ABG program changed their lives; thus, we did not use a predefined codebook as all codes were generated from the data themselves [55,56]. Meetings were held with members of the research team to triangulate coded text, produce summary matrices, and describe results organized by key overarching themes that emerged from the text. These themes included social and future well-being; the importance of school; connectedness to parents, culture, and community; and what participants learned in the ABG program.

## 3. Results

### 3.1. Quantitative Outcomes

We enrolled 394 participants in the study; 267 were randomized to receive ABG plus the control condition and 127 to the control condition alone. We collected baseline data from all participants, with the following retention rates: 72% at 6 months follow-up (*n* = 283; intervention: *n* = 192, control: *n* = 91); 75% at 12 months follow-up (*n* = 294; intervention: *n* = 202, control: *n* = 92); and 85% at 24 months follow-up (*n* = 335; intervention: *n* = 230, control: *n* = 105). There were no between-group differences in rates of retention/follow-up.

The average age of participants was 14.38 years and just over half were female (57.6%, *n* = 227). Over half of participants were cared for by both parents at baseline (59.6%, *n* = 235); 19.0% (*n* = 75) by one parent; and 20.4% (*n* = 84) by someone other than a parent. Participants were mobile: 44.5% (*n* = 175) had moved at least once in the past five years, and almost one-fifth had moved in the past year (19.9%, *n* = 78). Over one-third (36.3%) reported having low/very low food security. There were no between-group differences in baseline demographic characteristics.

#### 3.1.1. Entrepreneurship Knowledge

*Entrepreneurship Knowledge-Total:* Intervention participants had significantly improved total scores in entrepreneurship knowledge between baseline (13.93) and all follow-up times points (6 months: 16.63, *p* < 0.0001; 12 months: 16.94, *p* < 0.0001, 24 months: 18.01, *p* < 0.0001). Control participants had significantly improved scores in entrepreneurship knowledge between baseline (14.53) and two of the later follow-up time points (12 months: 15.88, *p* = 0.0093 and 24 months: 17.75, *p* < 0.0001). Gains in entrepreneurship knowledge were significantly greater for intervention vs. control participants at 6 months (16.63 vs. 15.51, *p* = 0.0071) and 12 months (16.94 vs. 15.88, *p* = 0.0083). (See Table 2).

*A: Personal Finance/Business Planning:* Intervention and control participants had nearly identical personal finance and business planning knowledge scores at baseline (8.86 vs. 8.87). Both groups made significant gains in their personal finance/business planning knowledge scores between baseline and each follow-up time point (ABG: baseline: 8.86; 6 months: 10.53; 12 months: 10.76; 24 months: 11.32; all *p*-values < 0.0001; Control: baseline: 8.87; 6 months: 9.90, *p* = 0.0039; 12 months: 10.05; *p* = 0.0008; 24 months: 11.06, *p* < 0.0001). There were no significant differences in trajectories over time between the two study groups. (See Table 2).

*Subscale B: Entrepreneurial/Business Development:* Intervention participants made significant gains in their entrepreneurial and business development knowledge between baseline (5.05) and all follow-up time points (6 months: 6.10, *p* < 0.0001; 12 months: 6.20, *p* < 0.0001; 24 months: 6.72, *p* < 0.0001). Control participants did not have significant improvements in their entrepreneurial/business development knowledge until 24 months follow-up (baseline: 5.67 vs. 24 months: 6.67, *p* = 0.0002). Gains made by intervention vs. control participants were significantly greater at all follow-up time points (6 months: *p* = 0.0015; 12 months: *p* = 0.0043; 24 months: *p* = 0.0428). (See Table 2).

#### 3.1.2. Economic Empowerment

*Expansion of Current Economic Abilities:* Intervention participants had significantly improved expansion in their current economic abilities between baseline (2.52) and all follow-up time points (6 months: 2.68, *p* < 0.0001; 12 months: 2.69, *p* < 0.0001; 24 months: 2.70, *p* < 0.0001). Control participants had improved scores from baseline only at the 24-month time point (2.47 vs. 2.68, *p* = 0.0001). There were no significant differences in trajectories over time between the two study groups. (See Table 2).

*Economic Agency and Participation:* Intervention participants had significantly improved economic agency and participation between baseline (2.26) and all follow-up time points (6 months: 2.72, *p* = 0.0151, 12 months: 2.73, *p* = 0.0043, 24 months: 2.77, *p* = 0.0001). Control participants had improved scores from baseline as well but not until the 24-month time point (2.62 vs. 2.75, *p* = 0.0123). There were no significant differences in trajectories over time between the two study groups. (See Table 2).

*Economic Confidence and Security:* Intervention participants reported improvements in feelings of economic confidence and security between baseline (2.47) and two follow-up time points: 12 months (2.56, *p* = 0.0027) and 24 months (2.54, *p* = 0.0128). Control participants did not have significant gains in economic confidence and security throughout the follow-up time period. There was a significant difference in trajectories of economic confidence and security between intervention and control participants at 12 months follow-up (2.56 vs. 2.43, *p* = 0.0063); otherwise, there were no significant differences in trajectories between groups at any time point. (See Table 2).

*Future Planning and Aspirations:* A significant gain in future planning and aspirations was observed for intervention participants between baseline and 24-months follow-up (3.04 vs. 3.16, *p* = 0.0012). Control participants did not have significant improvements in this domain. There were no significant differences in trajectories over time between the two study groups. (See Table 2).

*Intentions to Preserve Health:* A similar trend was observed for participant’s reported intentions to preserve health. Intervention participants had significant improvements in their intentions to preserve health between baseline and 24-months follow-up (2.87 vs. 2.94, *p* = 0.0059). Control participants did not have significant improvements in this domain. There were no significant differences in trajectories over time between the two study groups. (See Table 2).

#### 3.1.3. Connectedness

*Parents:* Participants who received ABG had a significant gain in their reported connectedness to parents between baseline and 24-months post-intervention (3.77 to 3.89, *p* = 0.0056). There were no other significant within-group or between-group changes in trajectories in connectedness to parents. (See Table 3).

*Mother:* ABG participants had steady significant improvements in connectedness to their mother between baseline (3.79) and all follow-up time points (6 months = 3.90, *p* = 0.05; 12 months = 3.95, *p* = 0.005; 24 months = 3.96, *p* = 0.001). Control group youth had a significant gain in connectedness to their mother but only between baseline and 24 months post (3.75 to 3.94, *p* = 0.02). There were no significant differences in trajectories over time between groups. (See Table 3).

*Father*: Similarly, intervention youth had steady significant improvements in connectedness to their father between baseline (3.39) and all follow-up time points (6 months = 3.51, *p* = 0.047; 12 months = 3.54, *p* = 0.0156; 24 months = 3.51, *p* = 0.037). Youth in the control group also experienced gains in connectedness to their father between baseline (3.21), 6 months (3.42, *p* = 0.027), and 12 months (3.44, *p* = 0.013). There were no significant differences in trajectories over time between the two study groups. (See Table 3).

*School and Teachers*: There was a significant between-group difference in gains made by the ABG group vs. control group between baseline and 6 months post in connectedness to teachers (ABG 3.42 to 3.50 vs. Control 3.44 to 3.33, *p* = 0.036). ABG participants also had a significant improvement in connectedness to teachers between baseline and 12 months follow-up (3.42 to 3.52, *p* = 0.045). There were no other significant within- or between-group changes in trajectories in connectedness to school or teachers. (See Table 3).

#### 3.1.4. Awareness of Connectedness

Participants receiving the ABG program made steady significant gains in their awareness of connectedness between baseline (33.04) and 6 months (34.86, *p* = 0.018), 12 months (35.13, *p* = 0.0056), and 24 months (36.12, *p* < 0.0001). Control group participants did not have a significant improvement in awareness of connectedness until 24 months follow-up (31.99 to 36.48, *p* < 0.0001). There were no significant differences in trajectories over time between the two study groups. (See Table 3).

#### 3.1.5. Hopelessness and Hopefulness

There were no significant within-group or between-group changes in trajectories in hopelessness or hopefulness at any follow-up time points. Changes in hopelessness were nearing significance between groups at the 12-month time point (4.01 to 3.91 vs. 3.81 to 4.36, *p* = 0.0798). (See Table 3).

### 3.2. Qualitative Outcomes

Quotes were selected by the study team that highlight how the ABG program was received by the participants and were sorted by specific domains: Psychosocial and future well-being; focus on school; connectedness to parents, culture, and community; and what youth learned in the ABG program. Illustrative quotes are presented for each of these aforementioned domains.

#### 3.2.1. Psychosocial and Future Well-Being

Focus group and interview participants described how the ABG program has improved their psychosocial well-being, more specifically their mood, being able to handle problems that come up in their lives, and staying out of trouble:


*“Well it got me through more hard things. Before we went to camp and everything, I was feeling depressed or sad. And I felt shy around a lot of the kids. And after camp and after all that stuff we’ve gone through with each other, I’ve gotten better. I’ve been more happy. I’ve been excited to go meet with the guys and the girls again, hang out. And it’s been having a good positive change on me. It’s been wonderful…. I’ve never been this happy in a long time.”*



*“I think ABG has pushed me more and has helped me learn that life has some ups and downs and you can’t just sit around and wait for something to happen. You got to make it happen yourself.”*



*“Before I usually get in a lot of trouble, but then when I started coming I never got in trouble as much as I used to.”*


Participants also described how they believe ABG has impacted their friends, and its potential to change their lives as well:


*“More kids are struggling to stay out of trouble. This [ABG] would really help them.”*



*“I think that it helps a lot with kids…sometimes people are depressed and have nothing to do and I think this really makes them come out and have something to do and think about college-wise. What are you going to do in the future? If your businesses are going to keep going.”*


Several participants described how the ABG program instilled in them a sense of pride, responsibility, and hope about their future:


*“It [ABG] gives you a sense of pride, and it teaches you responsibility…it helps you out with the future, and it teaches you that if you want something you have to work for it.”*



*“I am wanting to succeed more. I want to learn more so I can do better in my future. The program changed the way I see things now.…I see this opportunity like, ‘You know what? This could help you out in the future.’ It can help me advance and evolve.”*



*“It has helped me understand how the world works and how hard life can get. It helped me learn some ways that can help me in the future by creating my own business, that there’s not always just one option. You have so many options in the world that you can do. You can do so many things and you can’t just be shy. If you like it, do it. If you don’t, you can change it. There are so many things that you can do. So many things you can be.”*


#### 3.2.2. Focus on School

Numerous participants described how they changed their attitudes about school, more specifically that the ABG program helped them to focus on the importance of school and to make the connection that doing well in school will help them achieve their goals:


*“It [ABG] has helped a lot. Last year I was more focused on just messing around, having a good time during high school. But once I had to do my own business during the summer, I started having to work, it matured me. When I matured, it made me think of my future and how high school is the last step before you’re actually in the future. It made me focus on school more. It made me focus on just succeeding more.”*



*“Business has helped me focus more in school. Before I was like, ‘Oh, it’s school, okay, whatever.’ Now it’s like, ‘Okay, I’ve got to do this if I want to go into this business, or that one or that one,’ it’s made me push myself a little bit harder in school.”*



*“I have been doing better at school and more focused and I am achieving more.”*



*“Since I started coming, I didn’t really care about school, but since they’ve been teaching us how to run our own business, I want to go to school to teach the younger kids how to run their own business.”*


#### 3.2.3. Connectedness to Parents, Culture, and Community

Youth described how the ABG program has enhanced their connection to positive peers and resources in the community:


*“I made new friends at camp. And I didn’t have to get in a fight with anyone. Everyone was so nice there. And I could feel comfortable now with people that I don’t know, I feel way more comfortable with them now. I feel more trusted with them. I just love hanging out with the people, new people I never met.”*



*“It’s been amazing. I just hope more people at our school can think, ‘Hey, I’m not alone. I can go to this and I can hang out with the people that I don’t know, but I can meet.’ It’s been having a good thing on me. I just hope that one day, all of us can come together, and we can do this so we can get stronger.”*



*“It’s like a home away from home.”*


Participants also described how participating in the ABG program has created a deeper sense of trust and connectedness with their parents, through parents knowing they are engaged in a positive activity:


*“Because ever since I started coming here, they know I’m not doing anything crazy after school when I’m not home. They know I’m most likely here.”*



*“My parents trust me a lot more now.”*



*“My parents are happy that I’m in the business group. They think it’ll be a great learning experience and it has been. They think it’s an amazing program. They think it’s educational and something that they wish they would have had when they were young.”*


In addition, Apache youth described how the ABG program taught them more about their own culture, and instilled a deeper connection to their heritage:


*“The business group had workshops where we learned more about our own heritage and our own culture. I think ABG has really helped us learn more about who we are. The Apache culture is an amazing culture, I’m glad to be a part of it.”*



*“It’s been teaching us about what our culture was like and how we had to fight for it.”*


#### 3.2.4. What Youth Learned

In terms of skills the youth described learning in the ABG program, they spoke about those required to successfully manage a business ranging from financial literacy to how to interact with people:


*“I learned how to finance my own money and to be frugal with it.”*



*“I learned how to save money and how to use it to get things I need.”*



*“I learned I don’t want to waste money on little things that I don’t really need.”*



*“I learned how business works and how much time it takes to create a certain business.”*



*“I learned people skills. You learn how to interact with different people and how to speak in public.”*



*“You learn how to interact and you learn how to cooperate. When you’re talking to somebody or when you’re dealing with people you have to be calm, you have to be patient.”*


## 4. Discussion

This paper reports the effects of the ABG program on entrepreneurial, economic, and social well-being outcomes from both quantitative and qualitative data collected with Apache youth in a rural reservation. The reported results and discussion aim to advance knowledge from a rigorous evaluation of a youth entrepreneurship education program on assets and resources hypothesized as important drivers of health and well-being in this and other low-resource populations.

Intervention youth had gains in personal finance and business planning entrepreneurship knowledge that were significantly greater than gains made in the control group, and which persisted through 24 months of follow-up. There was a similar trend seen among intervention youth’s entrepreneurial and business development knowledge, which showed long-term statistically significant gains compared with youth in the control group. These quantitative findings are amplified by the voices of youth who described experiencing these entrepreneurship knowledge improvements in their own words, specifically how it grew their confidence in both handling money as well as their social skills. Thus, improvements in entrepreneurship knowledge may also portend increased mastery, independence, and self-efficacy, which are established protective factors against substance use and risks for suicide [24,25,29].

Youth who received the ABG program also had statistically significant increases in economic agency and participation as well as economic confidence and security between baseline and all follow-up time points. Control group youth did not experience significant gains in these economic empowerment outcomes until 24-months follow-up. While between-group differences in these outcomes were not significantly different at any follow-up time point, that intervention youth saw improvements in these domains in the short-term vs. control youth who did not see improvement until two-years later, is worth noting, as assets and resources acquired earlier in adolescence may buffer youth from multiple risks throughout seminal years of development [16,19,20]. In addition, youth who received ABG had significant long-term improvements in their future planning and aspirations and intentions to preserve health at the 24-month time point. This change is echoed by their voices elucidated in the qualitative data describing how participating in the ABG program changed their outlook, attitudes, and hopes for the future. Hope and optimism, in particular, have been identified as protective factors against suicide for Native youth; thus, bolstering these feelings through entrepreneurship education may be particularly relevant for Native communities [27].

Our findings regarding improvements in economic abilities and economic confidence among participants who received the ABG program could signify important contributions of the program in providing culturally meaningful income-earning opportunities for youth with limited prior employment or business experience. These impacts are further delineated in the words of the participating youth, who endorsed greater connection to school and parents and less frequent engagement in fighting and other nonproductive activities. Greater connectedness to school and family have been previously cited as promoting resilience, being commensurate with a positive youth development approach, and also direct buffers to substance use and suicide attempts among Native youth [35,36,37,38,39,40]. Linking entrepreneurial expansion to school, parents, and community connectedness may have substantive effects on positive health outcomes among Native youth. Although more research is needed in this area, our results underscore the promise of entrepreneurship education programs, such as ABG, for working upstream to prevent multiple risks for Native adolescents during a critical period of development. Our study did not find significant quantitative changes in ABG youth’s hopefulness or hopelessness, suggesting that more research is needed in this domain. The qualitative data, however, reflected increased positive thinking; it may be that the quantitative hope-related scales we utilized were not relevant to participants. Future economic-strengthening interventions may benefit from integrated health education or support to participants for developing short- and long-term financial goals.

### 4.1. Positive Unintended Consequences of the ABG program

While the ABG Café and Marketplace was originally envisioned as a space to provide opportunities for ABG participants to engage in experiential learning, the Café has grown over the study period to be a profitable healthy foods café, and perhaps the first focused on youth employment in rural reservations. It currently supports seven full-time employees, two of whom are ABG program graduates. The Café also spawned the ABG Food Truck that employs three full-time adult employees and has allowed the Café to expand its catering and special events business. Likewise, the Marketplace has grown into a local hub for emerging community artisans to sell goods and receive help from ABG staff to start new businesses. To date, the Marketplace has bought and sold products from over 300 adolescent and adult entrepreneurs from the Apache community. The Café and Marketplace are now a vital gathering place and community-based economic asset for this rural east Arizona tribal community.

As mentioned, the Café and Marketplace were inclusive and open to all community members; it follows that in a small community, such as the White Mountain Apache, youth in the control program may have frequented these businesses and interacted with intervention youth, as well as ABG program facilitators and community mentors. Thus, the potential for diffusion of ABG intervention effects in this tight-knit community are worth additional consideration and exploration. In addition, before the completion of follow-up data collection, the main local high school contracted ABG staff to teach the ABG curriculum to all incoming freshmen. These students may also be diffusing program content to other students, family, and community members. Future research should utilize social network tools to determine the diffusion of the ABG intervention’s impact and which specific components reached the highest risk youth [21].

### 4.2. Limitations

The decision to use a 2:1 randomization scheme decreased our power to detect statistically significant differences within the control group and between intervention and control groups. However, all trends favored the intervention group. The randomization choice was guided by our Apache CAB to afford as many Apache youth as possible to receive the program. As described, neither research staff nor study participants were blinded to randomization status; thus, results may to some degree be impacted by response bias. We did not have the resources to blind both research staff and study participants to randomization status. This paper presents the results of an intent-to-treat analysis; a sensitivity analysis should be done next to examine the impacts of ABG among youth who completed surveys at each follow-up time point. Though all assessments were selected for past use with Native youth and extensively pilot tested and edited prior to utilization, that we did not see the same quantitative impacts as demonstrated in the qualitative data, particularly with regard to some of the social well-being constructs, suggests additional work is needed to identify valid and reliable measures for use in this population. Additionally, the results may not be generalizable to other diverse tribal populations as data were collected with one tribal community, though many reservation communities endure similar health and economic disparities. In addition, potential diffusion effects within the community, and specifically to youth participating in the control group, are worth additional research and examination. Finally, as previously mentioned, we were not able to reach saturation with our qualitative data collection. However, the benefit of conducting a qualitative descriptive approach is that we were able to stay close to the data and surface of words to illustrate the experience of participants in the ABG program; with this as our goal, a qualitative descriptive approach is often touted as the method of choice [56].

## 5. Conclusions

The first rigorously evaluated youth entrepreneurship program designed for and by a Native American community appears to improve entrepreneurship, economic, and social well-being outcomes for participants, and potentially their peers and parents. Significant quantitative differences were observed between intervention and control participants in entrepreneurship knowledge at 6- and 12-months follow-up; entrepreneurship and business knowledge at 6-, 12-, and 24-months follow-up; economic confidence and security at 12-months follow-up; and connectedness to teachers at 6-months follow-up. Qualitative data illuminated meaningful improvements in youth’s psychosocial and future well-being, focus on school, and connectedness to their parents, culture, and community. The observed effects on entrepreneurship knowledge, economic empowerment, and connectedness, plus the experiences and changes as described by the youth themselves, demonstrates how a strength-based youth entrepreneurship intervention focused on developing assets and resources may be an innovative approach to dually address health and economic disparities endured in Native American communities. Future research should evaluate the implementation of ABG in other Native American and similarly under-resourced communities disproportionately impacted by public health disparities.

## Figures and Tables

**Table 1 ijerph-17-02383-t001:** Description of the outcome variables by category.

Variable	Number of Items	Response Options	Alpha
**Entrepreneurship Knowledge and Economic Empowerment**			
Entrepreneurship Knowledge: Total Subscale A: Personal finance/business planning Subscale B: Entrepreneurial/business development	31 18 13	Multiple Choice	0.7407
Economic Empowerment			
Expansion of current economic abilities	11	Likert (Range: 1–4)	0.8250
Economic agency and participation	7	Likert (Range: 1–4)	0.7310
Economic confidence and security	7	Likert (Range: 1–4)	0.6791
Future planning and aspirations	11	Likert (Range: 1–4)	0.9010
Intentions to preserve health	14	Likert (Range: 1–4)	0.7882
**Social Well-Being Outcomes**			
Connectedness			
Parents	6	Likert (Range: 1–5)	0.7223
Mother	5	Likert (Range: 1–5)	0.7414
Father	5	Likert (Range: 1–5)	0.7767
School	6	Likert (Range: 1–5)	0.8023
Teachers	6	Likert (Range: 1–5)	0.7337
Awareness of Connectedness	12	Likert (Range: 0–5)	0.8434
Hopelessness	16	True/False	0.6982
Hopefulness	13	Likert (Range: 1–5)	0.8149

Notes: This tables describes the outcome variables used in the ITT analysis including variable name, number of items comprising the variable, response options and alpha value for this sample.

**Table 2 ijerph-17-02383-t002:** Entrepreneurship knowledge and economic empowerment ^1^.

	*n brk* (Int; Cont)	Intervention	*p*-Value (within Group) ^2^	Control	*p*-value (within Group) ^3^	Trajectory Difference (Cont-Int)	*p*-Value (btwn Group Trajectory) ^4^
**Entrepreneurship Knowledge Total**							
**Baseline**	164; 77	13.93 (0.37)	Ref	14.53 (0.54)	Ref	Ref	
**6-Months**	191; 91	16.63 (0.36)	<0.0001	15.51 (0.52)	0.0647	−1.73 (0.64)	0.0071
**12-Months**	202; 91	16.94 (0.35)	<0.0001	15.88 (0.52)	0.0093	−1.66 (0.63)	0.0083
**24-Months**	230; 105	18.01 (0.34)	<0.0001	17.75 (0.50)	<0.0001	−0.87 (0.60)	0.1464
**Entrepreneurship Knowledge Subscale A: Personal Finance and Business Planning**							
**Baseline**	164; 77	8.86 (0.24)	Ref	8.87 (0.35)	Ref	Ref	
**6-Months**	191; 91	10.53 (0.23)	<0.0001	9.90 (0.33)	0.0039	−0.65 (0.43)	0.1328
**12-Months**	202; 91	10.76 (0.22)	<0.0001	10.05 (0.33)	0.0008	−0.73 (0.42)	0.0845
**24-Months**	230; 105	11.32 (0.21)	<0.0001	11.06 (0.32)	<0.0001	−0.27 (0.40)	0.5046
**Entrepreneurship Knowledge Subscale B: Entrepreneurial and Business Development**							
**Baseline**	164; 77	5.05 (0.18)	Ref	5.67 (0.27)	Ref	Ref	
**6-Months**	191; 91	6.10 (0.17)	<0.0001	5.60 (0.25)	0.8109	−1.12 (0.35)	0.0015
**12-Months**	202; 91	6.20 (0.17)	<0.0001	5.83 (0.25)	0.5738	−0.99 (0.35)	0.0043
**24-Months**	230; 105	6.72 (0.16)	<0.0001	6.67 (0.24)	0.0002	−0.67 (0.33)	0.0428
**Expansion of Current Economic Abilities**							
**Baseline**	266; 126	2.52 (0.03)	Ref	2.47 (0.04)	Ref	Ref	
**6-Months**	190; 90	2.68 (0.04)	<0.0001	2.55 (0.05)	0.1699	−0.08 (0.07)	0.2799
**12-Months**	200; 91	2.69 (0.03)	<0.0001	2.57 (0.05)	0.0772	−0.06 (0.07)	0.3722
**24-Months**	230; 105	2.70 (0.03)	<0.0001	2.68 (0.05)	0.0001	0.04 (0.07)	0.5777
**Economic Agency and Participation**							
**Baseline**	266; 126	2.62 (0.03)	Ref	2.62 (0.04)	Ref	Ref	
**6-Months**	190; 90	2.72 (0.03)	0.0151	2.71 (0.05)	0.1250	−0.01 (0.07)	0.9081
**12-Months**	201; 91	2.73 (0.03)	0.0043	2.71 (0.05)	0.0948	−0.01 (0.07)	0.8271
**24-Months**	230; 105	2.77 (0.03)	<0.0001	2.75 (0.05)	0.0123	−0.01 (0.07)	0.8555
**Economic Confidence and Security**							
**Baseline**	265; 123	2.47 (0.02)	Ref	2.49 (0.03)	Ref	Ref	
**6-Months**	189; 89	2.52 (0.03)	0.1158	2.48 (0.04)	0.8322	−0.06 (0.05)	0.2876
**12-Months**	201; 91	2.56 (0.03)	0.0027	2.43 (0.04)	0.2037	−0.14 (0.05)	0.0063
**24-Months**	230; 105	2.54 (0.02)	0.0128	2.51 (0.04)	0.6586	−0.05 (0.05)	0.3023
**Future Planning and Aspirations**							
**Baseline**	266; 126	3.04 (0.03)	Ref	3.03 (0.05)	Ref	Ref	
**6-Months**	190; 89	3.11 (0.04)	0.0716	3.09 (0.05)	0.2887	−0.01 (0.07)	0.8849
**12-Months**	200; 90	3.09 (0.04)	0.1483	2.99 (0.05)	0.4858	−0.10 (0.07)	0.1656
**24-Months**	230; 105	3.16 (0.03)	0.0012	3.05 (0.05)	0.7260	−0.10 (0.07)	0.1268
**Intentions to Preserve Health**							
**Baseline**	267; 126	2.87 (0.02)	Ref	2.83 (0.03)	Ref	Ref	
**6-Months**	190; 88	2.91 (0.02)	0.1787	2.88 (0.04)	0.1808	0.02 (0.04)	0.7286
**12-Months**	200; 90	2.91 (0.02)	0.1363	2.82 (0.04)	0.8791	−0.04 (0.04)	0.3376
**24-Months**	229; 105	2.94 (0.02)	0.0059	2.88 (0.03)	0.1685	−0.02 (0.04)	0.6823

Notes: This table shows within and between groups differences in entrepreneurship knowledge and economic empowerment outcomes among participants in the intervention and control groups. ^1^ All regression models included study group, time point, an interaction term between study group and time point, and were adjusted for repeated measures. ^2^ Within-group comparison to baseline value: intervention. ^3^ Within-group comparison to baseline value: control. ^4^ Between-group comparison of trajectory from baseline.

**Table 3 ijerph-17-02383-t003:** Social well-being outcomes ^1^.

	*N brk* (Int; Cont)	Intervention	*p*-Value (within Group) ^2^	Control	*p*-Value (within Group) ^3^	Trajectory Difference (Cont-Int)	*p*-Value (btwn Group Trajectory) ^4^
**Connectedness-Parents**							
**Baseline**	267; 127	3.77 (0.04)	Ref	3.78 (0.06)	Ref	Ref	
**6-Months**	190; 91	3.85 (0.05)	0.0808	3.81 (0.07)	0.6494	−0.05 (0.09)	0.5359
**12-Months**	202; 91	3.82 (0.05)	0.2329	3.73 (0.07)	0.5468	−0.10 (0.09)	0.2433
**24-Months**	230; 105	3.89 (0.04)	0.0056	3.83 (0.07)	0.4316	−0.07 (0.08)	0.3642
**Connectedness-Mother**							
**Baseline**	267; 127	3.79 (0.05)	Ref	3.75 (0.07)	Ref	Ref	
**6-Months**	188; 91	3.90 (0.05)	0.0502	3.94 (0.08)	0.0221	0.08 (0.10)	0.4453
**12-Months**	200; 89	3.95 (0.05)	0.0051	3.81 (0.08)	0.5173	−0.10 (0.10)	0.3052
**24-Months**	226; 103	3.96 (0.05)	0.0014	3.94 (0.08)	0.0202	0.01 (0.10)	0.8998
**Connectedness-Father**							
**Baseline**	253; 118	3.39 (0.06)	Ref	3.21 (0.09)	Ref	Ref	
**6-Months**	181; 85	3.51 (0.07)	0.0472	3.42 (0.10)	0.0272	0.08 (0.11)	0.4802
**12-Months**	189; 85	3.54 (0.07)	0.0156	3.44 (0.10)	0.0132	0.08 (0.11)	0.4788
**24-Months**	203; 94	3.51 (0.07)	0.0370	3.36 (0.10)	0.0924	0.02 (0.11)	0.8216
**Connectedness-School**							
**Baseline**	267; 127	3.58 (0.04)	Ref	3.57 (0.06)	Ref	Ref	
**6-Months**	190; 91	3.42 (0.05)	0.0010	3.47 (0.07)	0.1776	0.07 (0.09)	0.4448
**12-Months**	202; 91	3.50 (0.05)	0.0845	3.45 (0.07)	0.0950	−0.04 (0.09)	0.6762
**24-Months**	230; 105	3.42 (0.05)	0.0004	3.44(0.07)	0.0447	0.03 (0.08)	0.7318
**Connectedness-Teachers**							
**Baseline**	267; 127	3.42 (0.04)	Ref	3.44 (0.07)	Ref	Ref	
**6-Months**	190; 91	3.50 (0.05)	0.1546	3.33 (0.07)	0.1158	−0.19 (0.09)	0.0355
**12-Months**	202; 91	3.52 (0.05)	0.0448	3.42 (0.07)	0.7208	−0.13 (0.09)	0.1558
**24-Months**	230; 105	3.50 (0.05)	0.0912	3.47 (0.07)	0.7581	−0.06 (0.09)	0.4870
**Awareness of Connectedness**							
**Baseline**	267; 127	33.04 (0.67)	Ref	31.99 (0.97)	Ref	Ref	
**6-Months**	190; 91	34.86 (0.75)	0.0180	33.51 (1.09)	0.1697	−0.29 (1.35)	0.8293
**12-Months**	202; 91	35.13 (0.74)	0.0056	33.01 (1.09)	0.3557	−1.06 (1.34)	0.4316
**24-Months**	229; 104	36.12 (0.71)	<0.0001	36.48 (1.04)	<0.0001	1.42 (1.28)	0.2659
**Hopelessness**							
**Baseline**	267; 127	4.01 (0.17)	Ref	3.81 (0.25)	Ref	Ref	
**6-Months**	190; 91	4.11 (0.20)	0.6185	4.01 (0.29)	0.5155	0.09 (0.37)	0.8013
**12-Months**	202; 91	3.91 (0.19)	0.6301	4.36 (0.29)	0.0735	0.65 (0.37)	0.0798
**24-Months**	230; 105	3.79 (0.18)	0.2684	3.97 (0.27)	0.5843	0.38 (0.35)	0.2825
**Hopefulness**							
**Baseline**	267; 126	3.80 (0.03)	Ref	3.73 (0.05)	Ref	Ref	
**6-Months**	189; 91	3.77 (0.04)	0.3694	3.79 (0.06)	0.2621	0.10 (0.07)	0.1520
**12-Months**	202; 91	3.76 (0.04)	0.2581	3.75 (0.06)	0.6595	0.07 (0.07)	0.3184
**24-Months**	230; 105	3.76 (0.04)	0.2518	3.70 (0.05)	0.6747	0.02 (0.07)	0.7668

Notes: This table shows within- and between-group differences in psychosocial outcomes among participants in the intervention and control groups. ^1^ All regression models included study group, time point, an interaction term between study group and time point, and were adjusted for repeated measures. ^2^ Within-group comparison to baseline value: intervention. ^3^ Within-group comparison to baseline value: control. ^4^ Between-group comparison of trajectory from baseline.

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
