# Peer review of "Entrepreneurial, Economic, and Social Well-Being Outcomes from an RCT of a Youth Entrepreneurship Education Intervention among Native American Adolescents"

_ijerph, 2020, doi:10.3390/ijerph17072383_

Round 1

Reviewer 1 Report

The research is very interesting and the research design is well developed.

However, the methodology approach should be more developed, I think that a figure representing all the process would make it more clear, as there are several types of techniques present in this study.

The concepts should be better explained in the introduction, which should have also a more deepen literature review to support the empirical study.

The conclusions should be more specific linking the findings to the literature review. 

Author Response

Reviewer #1

The research is very interesting and the research design is well developed.

Thank you!

However, the methodology approach should be more developed, I think that a figure representing all the process would make it more clear, as there are several types of techniques present in this study.

We have previously published a paper describing our methodological approach which includes a figure representing the process/study flow. We have indicated where readers can find this information in line 37 (references #41, #42).

The concepts should be better explained in the introduction, which should have also a more deepen literature review to support the empirical study.

We have previously published two papers that include a comprehensive literature review and a deeper explanation of the constructs examined in this study. So as not to duplicate previously published material, we refer the reviewer to publications #41 and #42 from our study team.

The conclusions should be more specific linking the findings to the literature review.

We added more specific findings with linkage to references from the Introduction on lines 482-485, 490, 497-499, and 509-512.

Reviewer 2 Report

The topic and issues are interesting and worth publishing. 

An extensive discussion of the results were presented in section 4, discussion. However, a restate of the main points of evidence of the research paper and future research recommendations would be valuable in the conclusion section. 

Author Response

Reviewer #2

An extensive discussion of the results were presented in section 4, discussion. However, a restate of the main points of evidence of the research paper and future research recommendations would be valuable in the conclusion section.

We have restated the main points of the research paper on lines 558-564 and added future research recommendations to lines 568-570.

Reviewer 3 Report

This paper is to provide how a Youth Entrepreneurship Education Intervention can be applied on the Native American adolescents (who are members of the White Mountain Apache Tribe, Apache). The quantitative and qualitative approaches are used for collecting analyzed data from Arrowhead Business Group (ABG) program. And the analyzed results shown that the serval researched purposes have been clearly approved on entrepreneurial, economic and social well-being outcomes.

The paper has strongly provided a new way for enhancing on the Youth Entrepreneurship Education. Especially, this paper is closed touch practical experience and learning. Form the mentions of Arrowhead Café and Marketplace, the paper has delivered their efforts on the east Arizona tribal community. It shows that ABG program is not only a concept but also can be applied in practical operations.

However, this paper has a little bit problem should be consideration. The first one is the qualitative methodology. Based the most contents are descripted on quantitative analysis, thus the qualitative analysis is less descripted and not clearly (see line 265 to 273). The second one is more importantly, which is this paper has no theorical reviews to support their idea, although, this paper is well-writing, but what is the theorical foundations of this paper are difficult to find.

According to my personal opinions on two suggested points, this paper is encouraged to do some critical revises, then can be more suitable to publish on the International Journal of Environmental Research and Public Health.

Author Response

Reviewer #3:

However, this paper has a little bit problem should be consideration. The first one is the qualitative methodology. Based the most contents are descripted on quantitative analysis, thus the qualitative analysis is less descripted and not clearly (see line 265 to 273).

On lines 213 to 226 we describe the qualitative methodology that was utilized. This includes the method of data collection (focus groups and in-depth interviews), sampling method (purposeful), sample size (2 focus groups and 8 interviews), location of data collection (local study office), facilitators (Apache research staff), data collection tool (semi-structured interview guides), and open-ended questions included in the interview guides. This is the text currently included in the manuscript.

To complement and extend the quantitative data, we conducted focus group discussions and qualitative interviews with a subset of youth who participated in the ABG intervention to better understand how they understood and digested curriculum content, and potential mechanisms of intervention-related change. Participants were purposively sampled after completion of the ABG program. We conducted two focus groups with mixed-sex groups of youth and eight in-depth interviews with individual youth. Focus groups and in-depth interviews were conducted at our local study office by Apache research staff using semi-structured interview guides. Interview guides asked participants what they liked or disliked about the ABG program, how participating in the program has changed their lives, how their relationships have changed as a result of participating in the program, how they view Apache culture, and specifically their opinions regarding Apache values around entrepreneurship. Qualitative data were collected to enhance understanding of how the program was being received by the participants in terms of their economic, entrepreneurship and social well-being outcomes. We did not have resources to reach data saturation, but were able to gather consistent, rich illustrative quotes from the voices of participating Apache youth.

On lines 265-275 we describe the qualitative analysis methods used. This includes audio taping and transcription, use of two independent coders and a directed method of content analysis. We have added detail to this section including not using a predefined codebook (line 270), and how coded text was interpreted and organized by the study team (lines 271-273).

The second one is more importantly, which is this paper has no theorical reviews to support their idea, although, this paper is well-writing, but what is the theorical foundations of this paper are difficult to find.

The following is text pulled from our publication (#41 in reference list) describing the theoretical foundation for this paper. We kindly refer readers to this paper for further reference:

“This research to rigorously evaluate the ABG program with longitudinal follow-up seeks to lend momentum to shift the AI behavioral and mental health prevention research paradigm from a focus on risk reduction to protective factor promotion. Other research has indicated that protective versus risk-reduction approaches may be more effective in preventing behavioral and mental health risks among AI youth (see Borowsky, Resnick, Ireland, & Blum, 1999). Our intervention design and outcome evaluation is informed by qualitative and quantitative data gathered with Apache adolescents in previous studies conducted over nearly 20 years through our tribal-academic research partnership. Through this research we learned that 1) low educational achievement and school dropout, 2) hopelessness about the future, and 3) negative peer influences and activities are all significant risk factors for Apache youth substance use, suicide, and other high risk behaviors (see Barlow et al., 2012; Cwik et al., 2015, 2017, 2018; Tingey et al., 2012, 2014, 2016, 2017).

These findings are consistent with Jessor’s (1991) “Problem-Behavior Theory,” a long-standing social‐psychological framework that explains the interactional person-environmental determinants affecting adolescent health, with a focus on alcohol and drug use and other problem behaviors (see Jessor, 2017). We used our community-engaged process which included: a) formation and guidance on program design from a Community Advisory Board (CAB), b) input from the Apache Elders Council, and c) intensive review and editing by Apache research staff and faculty to adapt Jessor’s three major systems of explanatory variables. These systems are: 1) the perceived-environment system, 2) the personality system, and 3) the behavior system to design a conceptual framework for our ABG intervention.

First, our community-engaged process directed a departure from Jessor’s model to focus on protective (vs. risk) factors that have potential to buffer Apache youth from school drop-out, unemployment, substance use, violence, and suicide. These protective factors are empirically supported and include: a) connection to Apache history and values through time spent with Apache Elders; b) teaching of essential life skills including decision-making, problem-solving, goal-setting, and self-care; c) fostering positive connections with peers and caring adults through healthy recreational and community-based activities; and d) hands-on learning opportunities including entrepreneurship education and business development, job training, mentored apprenticeships, and college guidance (see Borowsky et al., 1999; Cwik et al., 2017, 2018; Kenyon & Carter, 2011; Stiffman et al., 2007; Tingey et al., 2014, 2016, 2017).

Input from core Apache staff and faculty who were guided by the local CAB and Elders’ Council led to further modification of Jessor’s model to reflect the intra-personal domains that could be impacted within participating youth: cultural identity, self-care, social relationships, and opportunities. A final departure from Jessor’s model was to replace the original linear pathway illustration with a circular, dynamic flow model showing constant reciprocal exchanges between an individual and ABG’s targeted growth constructs (cultural identity, self-care, social relationships, and opportunities) and the way ABG seeks to engage community in a constant, evolving cycle to shape lifelong behavioral repertoires (see Figure below). The final ABG theoretical model is a deep structure cultural adaptation (see Okamoto, Kuli, Marsiglia, Holleran-Steiker, & Dustman, 2014), if not counter reaction, to Jessor’s original model that is ingrained with Apache knowledge and aspirations for ABG’s impact on youth development within a nurturing community context.”

(Figure is in attached file version of the response).
